# Neuronal gene expression in two generations of the marine parasitic worm, *Cryptocotyle lingua*

Oleg Tolstenkov [1✉], Marios Chatzigeorgiou [1✉] & Alexander Gorbushin [2]

Trematodes, or flukes, undergo intricate anatomical and behavioral transformations during their life cycle, yet the functional changes in their nervous system remain poorly understood. We investigated the molecular basis of nervous system function in *Cryptocotyle lingua*, a species of relevance for fisheries. Transcriptomic analysis revealed a streamlined molecular toolkit with the absence of key signaling pathways and ion channels. Notably, we observed the loss of nitric oxide synthase across the Platyhelminthes. Furthermore, we identified upregulated neuronal genes in dispersal larvae, including those involved in aminergic pathways, synaptic vesicle trafficking, TRPA channels, and surprisingly nitric oxide receptors. Using neuronal markers and in situ hybridization, we hypothesized their functional relevance to larval adaptations and host-finding strategies. Additionally, employing a behavior quantification toolkit, we assessed cercaria motility, facilitating further investigations into the behavior and physiology of parasitic flatworms. This study enhances our understanding of trematode neurobiology and provides insights for targeted antiparasitic strategies.

[1] Michael Sars Center, University of Bergen, Bergen, Norway. [2] Sechenov Institute of Evolutionary Physiology and Biochemistry, St Petersburg, Russia. ✉email: oleg.tolstenkov@uib.no; Marios.Chatzigeorgiou@uib.no

Trematodes, or flukes, cause disease in millions of people, impact animal health, and alter the functional organization of biological communities. To develop strategies to combat these severely detrimental effects, it is fundamental to understand their biology, especially that of the nervous system, through research on different model species across life stages[1]. Enabled by earlier work on the genomic and transcriptomic analyses of a few model trematode species of medical relevance, some major features of neuronal gene organization in trematodes have already been characterized[2,3]. However, limiting research to only a few trematode species is disadvantageous because trematodes are known to exhibit species-dependent host-pathogen interactions, which in turn give rise to highly variable mechanisms of host finding and infection. An urgent challenge in the field is to broaden the spectrum of trematode species that researchers work on. Thus, this work aims to contribute to the expansion of marine trematode models for neuroscience research, such as those that are not hazardous to humans, thus allowing for more quantitative and systematic experimentation in the laboratory setting.

Nervous system streamlining and plasticity are fundamental processes that shape neural function over time, enabling the nervous system to adapt to changing environmental demands. Trematodes have a highly complex life cycle that few animal models can match. For instance, the generational transition from the intramolluscan redia to the freely swimming cercaria comes with extensive anatomical and behavioral modifications, enabling the larva to locate and infect the next host in the complex water environment[4,5]. However, the functional changes that occur in the nervous system during this transition are not well understood.

The trematode species *Cryptocotyle lingua* (Creplin, 1825) (Digenea: Heterophyidae) is widespread in the North Atlantic Ocean, in European waters and the Northwest Atlantic (WoRMS database). It uses coastal birds as definitive hosts, where the parasites develop as hermaphroditic adults and produce eggs. A ciliated larval stage, called miracidia, hatches from the egg and infects sea snails (periwinkles, genus *Littorina*), where it develops into a sporocyst. The latter produces parthenogenetic daughter generations represented by rediae[6]. The rediae produce and release free-swimming, short-lived, and eye-spotted cercariae that demonstrate characteristic sensitivity to light and water turbulence stimuli[7,8]. These larvae infect downstream hosts (Fig. 1a), causing the common 'black spot disease' in several marine fish species, including the cod, which can cause loss to both the economy and the ecosystem[9]. Here, cercariae penetrate the fish skin, encyst in the hypodermis, and survive as metacercariae for a long time until the intermediate host is eaten by the definitive host. Aside from causing substantial problems for fisheries[10–13], *C. lingua* is often used in ecological[14,15], immunobiological[16], and behavior studies[17,18].

Despite the importance of the nervous system in the trematode life cycle progression, little is known about the molecular mechanisms underlying this process. To address this gap in knowledge, we conducted a transcriptomic analysis of the nervous system in cercariae and rediae of the fluke *C lingua*, focusing on genes differentially expressed between these two phases of the life cycle. To elucidate the molecular and behavioral basis of the promising model species *C. lingua* for further neurobiological studies, this work aims to gain insights into nervous system function in two stages of the life cycle. To establish *C. lingua* as an experimental model in functional neuroscience, this work has characterized the complexity of behavior of the cercaria stage using live behavioral recording combined with computer vision. Furthermore, to link these animal behaviors to neural function, nervous system development, and gene expression, we performed de novo transcriptome assembly of *C. lingua* across two life cycle stages, in combination with whole-mount antibody staining and in situ hybridization analysis of gene expression patterns, to identify important genes in neurotransmission that may play cercariae stage-specific functions.

## Results

**Dispersal larvae of *C. lingua* exhibit complex but quantifiable behavior.** Rediae are tissue parasites that proliferate and produce cercariae (Fig. 1a) in the host's circulatory system[6], gonads, and hemolymph lacunae[19]. The redia has a rudimentary nervous system[20], that is specifically adapted to the intramolluscan environment. In contrast, fully developed cercariae have a functional nervous system that allows them to sense and respond to environmental cues[5] (Fig. 1b). In our experiments, we could not qualify spontaneous slow wiggling of rediae in ASW as behavior (Supplementary Movie 1). We characterized the speed, timing of activity states, and path complexity of a limited sample of cercaria using computer vision and analysis toolkit[21,22] (see methods section for details). Cercariae of *C. lingua*, consistent with previous studies, exhibited intermittent swimming behavior (Supplementary Movie 2, Fig. 1c, and Supplementary Note 1, Supplementary Fig. 1 in Supplementary Information file). However, based on our analysis of path complexity during spontaneous locomotion, our data reveals previously uncharacterized patterns within burst-like activity in *C. lingua* cercariae (Supplementary Fig. 1). This suggests that their behavior may exhibit a higher degree of complexity and potentially operate in more dimensions than previously thought (Fig. 1ci–v).

**Deep sequencing of *C. lingua* transcriptome resulted in high-quality assembly and protein prediction.** We obtained 159.1 million strand-specific paired reads in all six libraries (21.1–23.9 million per each), which were used in the 'raw' transcriptome assembly. The fraction of the raw assembly, the 'curated assembly', was produced by filtering out non-coding Trinity clusters ('genes'), low-expressed genes, and few genes of xenogenic origin (the host *L. littorea*, bacteria, fungi, and human). Out of 12698 proteins, 6288 were annotated with a BLAST hit (E-value cut-off: 1E-25) and sequence descriptor. Predicted protein length varied from 89 to 8093 amino acids. BUSCO completeness assessments and protein statistics showed high assembly quality and protein prediction comparable to other trematode model species[23] (Fig. 1d, e).

**Gene expression patterns reveal redia's role in cercaria production and highlight cercariae's dispersal-related characteristics.** Our *C. lingua* transcriptome analysis revealed a high number of mobile elements such as LINE-1, PiggyBac, Tigger, Pogo transposable elements, and gag-pol polyproteins. However, for the scope of this study, these elements were excluded from our primary analyses. In the subset of 9338 protein-coding genes, 1355 (14.5%) were four-fold upregulated (FDR ≤ 0.05) in rediae, while only 296 (3.2%) in cercariae (Fig. 2a, b). Among top 20 stage-specific genes, several proteinases required for enzymatic digestion were presented in redia-specific DEGs, consistent with the active feeding of intramolluscan redia. The primary biological function of the redia stage is the production of dispersal larvae, the cercariae. Accordingly, of the 74 development-regulatory Homeobox genes, only six were upregulated in the cercariae. In contrast, 27 Homeobox genes showed specific upregulation during the redia stage. This suggests a potential homeobox signature for *C. lingua* embryonic development (Supplementary Data 1, Fig. 2b).

To gain insight into the stage-specific biological processes in the *C. lingua* expression profile, we conducted a Gene Ontology (GO) enrichment analysis on the DEG sets of rediae and

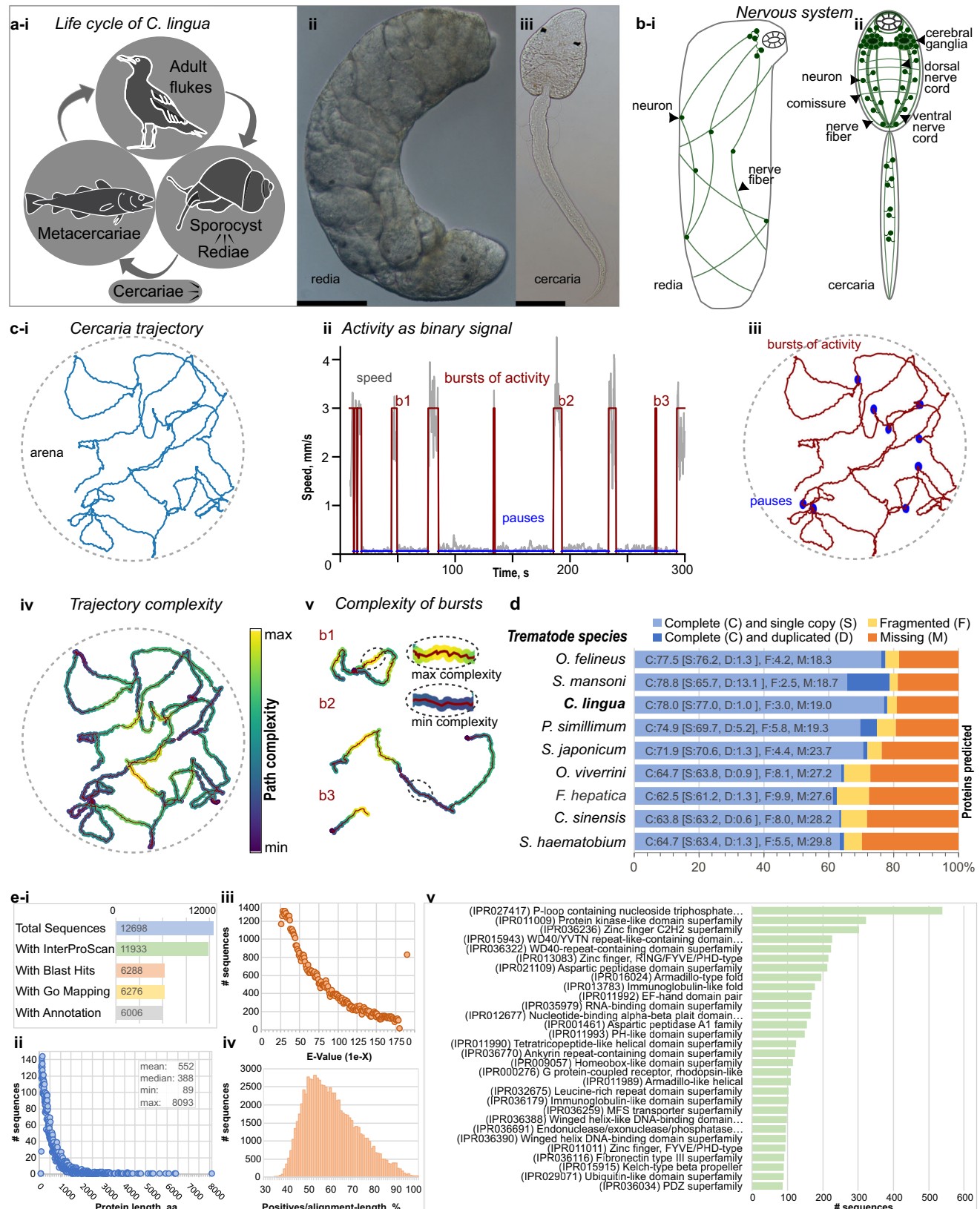

cercariae. The results were consistent with the expected biology of these life cycle stages: the parthenogenetically breeding generation (rediae) produces actively swimming larvae (cercariae). Rediae, which contain all stages of cercaria embryo development, expressed genes that control various cellular processes, including developmental processes and morphogenesis (Fig. 2c).

Additionally, redia-specific gene expression includes multicellular organismal processes that regulate digestion and nervous system functions. In contrast, mature cercariae were sampled after actively swimming for at least four hours after shedding, and GO analysis revealed a highly represented group of terms related to metabolic processes, including catabolic processes, primary,

**Fig. 1 Introduction in the model species *C. lingua*, transcriptome assembly and protein prediction. a** *C. lingua* life cycle (**i**), intramolluscan redia with developing cercariae embryos (**ii**) and free-swimming cercaria (**iii**), scale bar = 50 μm. **b** Nervous system increases anatomic complexity in cercaria (**ii**) compared to redia (**i**), cartoon based on the neuronal markers staining (see below). **c** Trajectories analysis showed complexity of cercaria behavior. Representative trajectory (**i**), traditional approach of cercaria behavior analysis reveals easy to quantify two states of activity (**ii**) that are plotted on trajectory (**iii**), and results of path complexity (entropy) analysis with complexity of trajectory with 1 second window (**iv**) and individual bursts of activity with specified parts of maximum and minimum path complexity (**v**). The plotted data depict a behavior of representative single cercaria four hours post-shedding. Refer to Supplementary Fig. 1 for additional trajectories and metric distributions and to Supplementary Movie 2 for video recordings of the behavior. **d** Summarized BUSCO (v4.1.4; Metazoa_odb10) benchmarking of protein arrays predicted in *C. lingua* transcriptome and in proteomes of eight other digenean species (*Schistosoma mansoni* (UP000008854), *S. japonicum* (UP000311919), *S. haematobium* (UP000471633), *Opisthorchis viverrine* (UP000054324), *O. felineus* (UP000308267), *Clonorchis sinensis* (UP000286415), *Fasciola hepatica* (UP000230066) and *Psilotrema simillimum*[23]. Bar charts show proportions of 954 BUSCO orthologs classified as complete, fragmented, and missing. **e** Descriptive statistics of proteins inferred from *C. lingua* transcriptome. (**i**) summary of protein statistics; (**ii**) length distribution of predicted proteins; (**iii**) E-value distribution of BLASTp hits in SwissProt database; (**iv**) protein sequence similarity to curated SwissProt prototypes; (**v**) top 30 InterProScan protein domain superfamilies. The data presented were derived from three independent biological replicates.

cellular, and small molecule metabolism (Fig. 2c). This finding is consistent with the dispersal function of endotrophic larvae, which use internal sources of energy for swimming and indicates the high metabolic rate in host-seeking larva cercaria. Notably, the rediae exhibited a different metabolic process, which was represented by protein and macromolecular metabolism (Fig. 2c).

To investigate the molecular-level function of the nervous system in *C. lingua*, we focused on 364 referenced neural proteins, identified and classified 226 homologous protein-encoding transcripts in the transcriptome of *C. lingua* (NCBI GenBank, accession numbers MW361065–MW361239). We performed a differential expression analysis of selected neuronal genes in redia and cercaria and characterized the domain architecture of the protein and the expression profile of each neuronal gene. We characterized transcripts involved in chemical and electric synaptic machinery, transmission with classical neurotransmitters and small gaseous messengers, reception machinery, and voltage-gated channels (Supplementary information file, Supplementary Note 2, Supplementary Fig. 2 and 3, Supplementary Data 2–9). Here we shortly characterize the sets of neuronal proteins and their expression between stages.

The biosynthesis of classical neurotransmitters (NT) and small gaseous messengers is a crucial step in neurotransmission, and the identification of corresponding enzymes in the transcriptome provides evidence for the existence of these pathways[24]. We confirmed the presence of biogenic amines pathways, acetylcholine glutamate, and transsulfuration pathway (Fig. 3a). The complete list of enzymes and more details can be found in the Supplementary information (Supplementary Note 2, Supplementary Data 2, Supplementary Fig. 2a).

A subset of genes were significantly upregulated in rediae, including ortholog of biogenic amine synthesis DDC, as well as enzymes that are not specific to the nervous system, such as cholinesterases and glutaminase. An equal-in-size subset of genes upregulated in cercaria included *CHAT* and biogenic amine synthesis genes *DBH*, *TDC*, and *TPH* (Fig. 3b, Supplementary Data 2).

We failed to find orthologs of several genes involved in neurotransmitter biosynthesis and metabolism, including tyramine beta-hydroxylase, histidine decarboxylase, histamine N-methyltransferase, and nitric oxide synthases (Supplementary Data 2).

The nitric oxide synthase (NOS) enzyme belongs to a family of CPR (cytochrome P450 Reductase) enzymes that contain both flavodoxin-like (FMN-binding) and FAD domains. This family includes NADPH-cytochrome P450 reductase (NCPR) and bacterial sulfite reductase[25]. In the *C. lingua* transcriptome, only NCPR (GeneBank: MW361191) is present, which is composed of four structural motifs: transmembrane helices, flavodoxin-like

domain, FAD-binding domain, and NAD-binding domain (Fig. 3a–iii). However, there are no transcripts in the assembly encoding protein with the NO_synthase (syn. NOS_oxigenase, NOSoxy) domain, which is the hallmark structural characteristic of all types of NOS[26]. Moreover, this N-terminal and functionally central domain is absent in all 33 Platyhelminthes genomes and 49 transcriptomes (NCBI TSA projects) available, including representatives of classes Trematoda, Monogenea, Cestoda, and free-living turbellarians *Macrostomum lignano* and *Schmidtea mediterranea*.

Consistent with the neurotransmitter systems our analysis revealed the presence of genes involved in vesicle loading, but no GABA transporters. A few of these genes were found to be upregulated in the rediae, and one gene in cercaria (Fig. 3b). For a more detailed list of homologs and expression profiles, please refer to Supplementary Data 3 and Supplementary Fig. 2a.

We have identified genes from all core groups[27] known for vesicle trafficking, release, and recycling, including 10 synapto-tagmins that act as calcium sensors[28], Rab proteins important for docking and tethering vesicles, proteins of the SNARE complex, synapsin, transmembrane adenosine triphosphatases, and other proteins. A detailed list of homologs and their domain structures can be found in Supplementary Data 4 and 5, and Supplementary Fig. 2a.

However, we did not find two chains of synuclein, a multifunctional protein that works as a chaperone in the SNARE complex, *Rab GTPase 3 A* genes, G-proteins that guide synaptic vesicles to active synaptic zones, or *Rab effector Noc2*, amphiphysin, parvalbumin alpha, and adhesion G protein-coupled receptors.

Of the 75 genes characterized in this subset, 15 were significantly upregulated in cercariae: including co-chaperones HSPA8 and PACSIN adapter protein, SNARE complex protein SNAPA, and zinc transporter SLC30A3. Another five differentially expressed genes were upregulated in rediae, including two synaptotagmins (*SYT2* and -*3*), which are rediae-stage specific.

Based on gene ontology annotation, over 150 putative G protein-coupled receptors (GPCRs) belonging to all major GPCR groups, and around 100 putative ligand-gated ion channels from three major groups, namely Cys-loop family, glutamate-activated cation channels, and ATP-gated ion channels[2], are present in the transcriptome. A selected set of putative receptor proteins was characterized based on reference genes with proven function, including ionotropic and metabotropic receptors, P2X purinor-eceptor, soluble guanylate cyclase subunits, some proteins engaged in the second-messenger cascade, and proteins regulating receptor functions.

From the neurotransmitter ligand-gated ion channels, we identified nine nicotinic acetylcholine receptors, orthologs of the

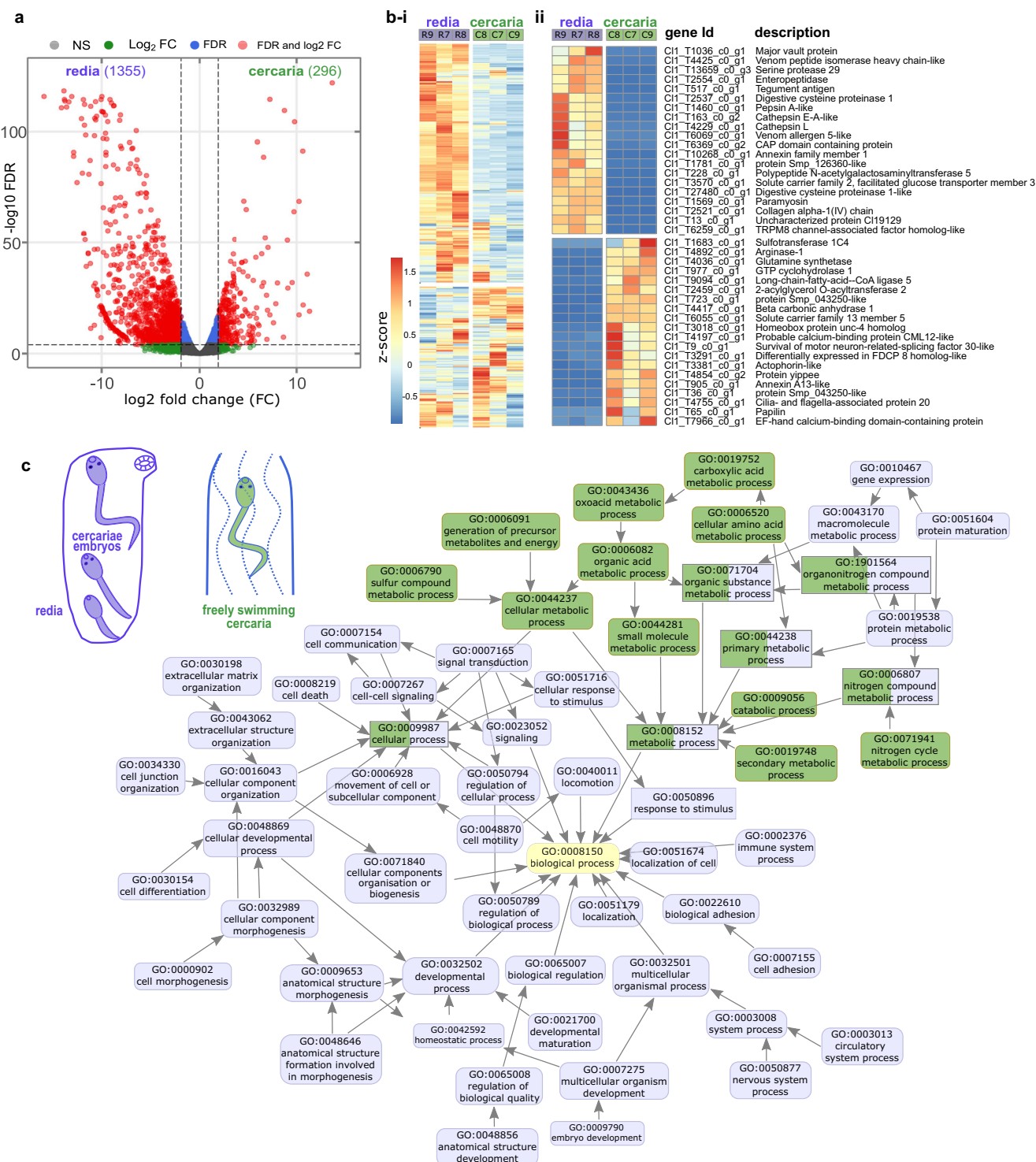

**Fig. 2 Differential expression analysis of *C. lingua* transcriptome (9338 unigenes) was consistent with the production and dispersal functions of redia and cercaria. a** Volcano plot of differentially expressed genes (DEGs) in rediae and cercariae. Numbers of stage-specific DEGs are shown in parentheses. The red dots indicate significantly regulated genes (*threshold*: logFC ≥ 2, FDR < 0.05), the blue and green dots indicate under-threshold genes, respectively, for FC and for FDR, the gray dots indicate non-significant differences. **b** Heatmap showing gene expression profiles (cross-sample normalized TMM values) in three redia and three cercaria samples (**i**). Red and blue indicate up- and down-regulated expression. The scale bar represents the row *Z* score. Top 20 proteins that were significantly down- and upregulated in cercariae (**ii**). **c** Combined graph of Gene ontology (GO) term enrichment analysis for differentially expressed genes in *C. lingua* rediae and cercariae. Only overrepresented (FDR < 0.05) functional annotations for Biological processes are displayed. Annotations are indicated with nodes colored by the developmental stage of the parasite. The data presented were derived from three independent biological replicates.

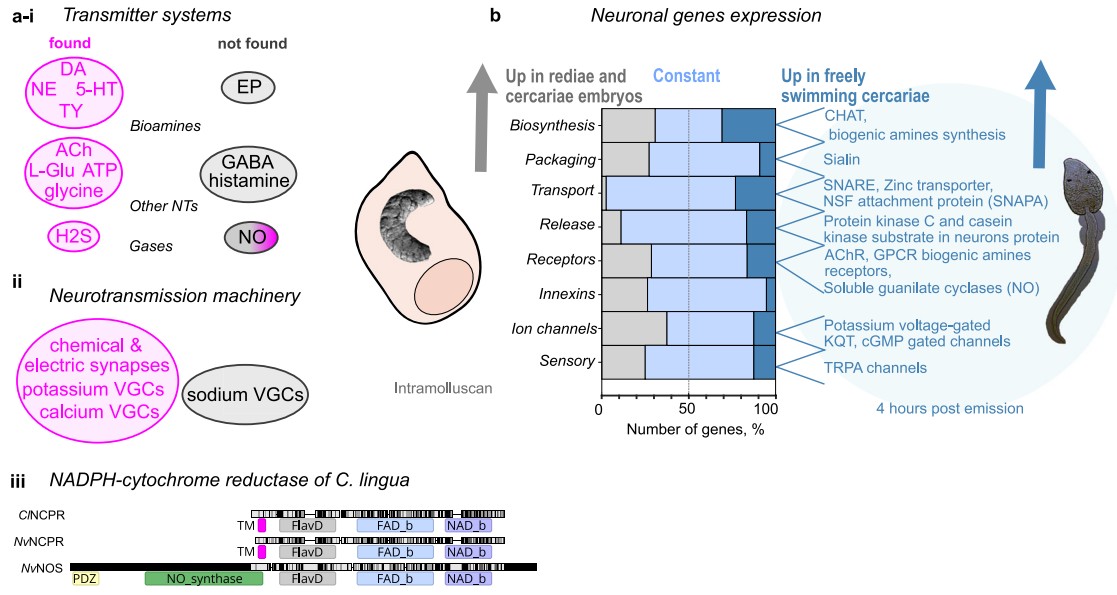

**Fig. 3 Neuronal proteins are not equally expressed in rediae and cercariae. a** Presence (magenta) and absence (gray) of homologs proteins related to neurotransmitter systems (**i**) and machinery of neurotransmission (**ii**). Based on enzymes involved in biosynthesis, specific proteins of neurotransmitter packaging, receptors, and metabolism we registered the neurotransmitters pathways for dopamine (DA), norepinephrine (NE), serotonin (5-HT), tyramine (TY), acetylcholine (Ach), glutamate (L-Glu), glycine, hydrogen sulfide (H2S), and adenosine triphosphate (ATP). We did not register homologs of proteins involved in epinephrine biosynthesis (EP), pathways of gamma-aminobutyric acid (GABA), histamine, and nitric oxide synthases (NOS), while the nitric oxide (NO) receptors soluble guanylate cyclases were present. (**ii**) Presence and absence of various voltage-gated channels (VGCs). The data presented were derived from three independent biological replicates. (**iii**) Domain alignment of *C. lingua* NADPH--cytochrome P450 reductase (*Cl*NCPR), closest protein to NOS in *C. lingua* transcriptome with two *Nematostella vectensis* enzymes from CPR family, the orthologous protein *Nv*NCPR (XP_032236207) and Nitric oxide synthase, brain isoform X1 *Nv*NOS (XP_032236179). PFAM domains PDZ, NO_synthase, Flavodoxin, FAD-, NAD-binding, and transmembrane helices (TM) are shown. **b** Representation of neuronal DEGs in two stages studied. Genes upregulated in rediae are shown in gray, constant in blue, and the genes upregulated in freely swimming larvae 4 hours after shedding are in dark blue. Percentage was calculated from the number of genes in each subgroup characterized. Differential expression was calculated for three paired biological replicates (See Methods section and Supplementary Data 2–9).

acetylcholine-gated chloride channels of *Schistosoma mansoni* ACC-1 and ACC-2, inhibitory chloride ion channel glycine receptors, glutamate receptors - cation channels kainite and N-methyl-D-aspartate (NMDA), and chloride channel. From metabotropic receptors, we identified orthologs of muscarinic acetylcholine receptors, dopamine receptors, serotonin receptors, octopamine-tyramine receptors, and orthologs of metabotropic glutamate receptor. We also classified two orthologs of P2X purinoreceptors and nitric oxide receptors soluble guanylate cyclases. A detailed list of homologs, their domain structures, and expression profiles can be found in Supplementary Data 6, and Supplementary Fig. 3a.

All the reception machinery genes were expressed at relatively low levels (Supplementary Data 6, Supplementary Fig. 3a). Transcripts of 27 genes showed equal abundance in rediae and cercariae, including acetylcholine gated chloride channel, nicotinic acetylcholine receptors, NMDA, and kainate receptors, metabotropic glutamate receptor, putative metabotropic receptors, putative biogenic amines receptors. One of the acetylcholine-gated chloride channels, putative serotonin receptor *HTR1A2*, putative GPCR1 receptors, and soluble guanylate cyclases were upregulated in free-swimming larvae. All other 14 DEGs of 49 characterized proteins were upregulated in rediae (Fig. 3b).

We failed to find GABA receptors in either *C. lingua* nor other Plathelminthes, however, structurally similar multi-pass membrane proteins, the glycine receptors, (GLRB, GLRA) were identified in *C. lingua*. We also did not find homologs of serotonin 5HT3 (Ionotropic), HTR2, 5, 6, 7 receptors, and

glutamate quisqualate (AMDA) receptors, adrenergic receptors, and P2Y receptors in our dataset.

We did not find sodium voltage-gated ion channels in the transcriptome of *C. lingua*, only a sodium leak channel non-selective protein (NALCN) that is orthologous to the voltage-independent, cation-non-selective NALCN found in mammals. This protein is permeable to sodium, potassium, and calcium ions (UniProt). Interestingly, the genomes of cestodes and free-living flatworms contain orthologs of the α-subunits forming the core of the voltage-gated sodium channels (SCN1A–SCN11A), while trematodes do not. We also characterized potassium voltage-gated channels, which are the most diverse group of the ion channel family. Our analysis included representatives of subfamily A, H, Shal, Shab KQT, TWIK, voltage-dependent calcium channels, SK, BK, and homologs of cGMP-gated cation channels. The domain structures and expression profiles are presented in Supplementary Data 7 and Supplementary Fig. 3c. In *S. mansoni* genome 41 putative channels were identified[2] but only few are characterized functionally[29]. The expression of voltage-gated ion channels in *C. lingua* was relatively low, two channels were upregulated in cercariae, and all other 10 differentially expressed genes were upregulated in rediae (see Fig. 3b).

In the transcriptome, innexins are represented by 16 genes (Supplementary Data 8 and Supplementary Fig. 3c). Only two differentially expressed innexins were upregulated in cercariae and six in rediae (Fig. 3b).

Proteins involved in sensory perception play a critical role in converting input stimuli into membrane potential changes in

sensory neurons. Our research has identified several orthologs, including transient receptor ion channels TRPA1-A18, a non-specific cation channel Piezo, and mechanosensory protein 2 mec-2. Additionally, we have identified several orthologs of light-absorbing opsins, specifically OPN1-4. Supplementary Data 9, Supplementary Note 2, and Supplementary Fig. 3d provide detailed domain structures and expression profiles.

Among the four opsins identified in the *C. lingua* transcriptome, OPN1 displays a conservative motif indicative of Gq-opsin families. Moreover, two highly conserved peptide amino acids histidine and proline, in the carboxy-terminal intra-cellular loop domain show significant (64–67%) identity to the characterized rhabdomeric (r) Gq-opsins of *S. haematobium* and *S. mansoni* (Smp_104210 and Sha_101185))[30], which have a suggested role in photokinesis[31]. Rhabdomeric opsins are known to attain color vision in invertebrates, with higher sensitivity to blue light[32]. Interestingly, the expression of *ClOPN1* is higher than other putative opsins (Supplementary Fig. 3d) in both studied stages, which corresponds to the experimental data on the predominant blue-light sensitivity of *C. lingua* cercaria[8]. Most of the differentially expressed genes (DEGs) in this subgroup were upregulated in cercariae developing in rediae, while three TRPA channels (*ClTRPA 9, 10,* and *16*) were upregulated in freely swimming larvae (Fig. 3b).

We did not find degenerins, mechanosensory abnormality protein 6, transmembrane channel-like proteins, mechanosensory transduction channel NOMPC, and amiloride-sensitive sodium channel subunits in the transcriptome of *C. lingua*.

**The HCR in situ hybridization of selected neuronal genes has revealed their expression in the nervous system.** Although most of the neuronal proteins were upregulated in rediae samples, only a few were upregulated in cercariae. To obtain greater insights into the function and expression patterns of these selected genes, we performed HCR in situ hybridization.

We proved that characterized neuronal genes are expressed in the nervous system of *C. lingua*. To do this, we used staining with neuronal marker antibodies (AB) and HCR in situ hybridization for the same transcripts in parallel. We were aware of the limitations of using commercial AB, which was not previously tested on trematode proteins. Nevertheless, we managed to compare the localization of AB staining with the results of in situ HCR based on the anatomy of the nervous system described earlier[33,34]. All three neuronal markers used: calcium sensor synaptotagmin, ubiquitous in the nervous system[35]; CHAT, enzyme essential for the production of acetylcholine, and serotonin or SERT—specific serotonin transporter; were localized or stained in the nervous system. They were localized both in the central nervous system, cerebral ganglia, commissure, and ventral nerve cords, as well as in peripheral neurons, nerve cords, and nerves of cercaria. Interestingly in contrast to AB staining, no neuronal markers expression was localized in the nervous system of redia itself, only in developing cercaria within redia (Fig. 4).

**Expression of genes upregulated in dispersal larva suggests their role in the nervous system maintenance and facilitating host finding.** The use of HCR in situ hybridization allowed for the quantification of mRNA levels[36] for three selected genes, namely *ClSYT1*, *ClCHAT*, and *ClSNAP*, providing additional evidence for our DEG analysis. We compared the fluorescence in the central nervous system between three samples, redia, developing cercaria, and freely swimming cercaria (Fig. 5a), and found the localization of neuronal markers in the nervous system of developing and freely swimming cercaria but not in redia. The

results of HCR quantification were generally consistent with our RNAseq data (Fig. 5b–d).

To further investigate the expression of a set of neuronal genes upregulated in dispersal larvae, we localized the transcripts of selected upregulated genes. Alpha-soluble NSF attachment protein *ClSNAPA* plays a role in the fusion of synaptic vesicles to the plasma membrane[37], while *ClPACSIN1* is involved in synaptic vesicle endocytosis[38]. We assumed that both transcripts should be localized in synapses, and as a result, expression was found to be distributed along the body of the cercaria with aggregation in the central nervous system (Fig. 5d, Supplementary Fig. 3). Putative serotonin receptor *ClHTR1A2*, on the other hand, was detected in the cerebral ganglia and in the neurons of the tail (Fig. 5e).

The transcripts of the receptor-activated non-selective cation channels *ClTRPA10* and *ClTRPA16*, homologs of polymodal TRPA1 with suggested temperature, chemical and mechanosensitivity[39], were found to be expressed differently within the aggregation of cilia in the area around and below the oral sucker for *TRPA10* (Fig. 5f), as well as at the surface of the tail and in the set of peripheral neurons on both sides of the body for *TRPA16* (Fig. 5g). Additionally, the soluble guanylate cyclases *ClGUCY1B1* and *ClGUCY1B2*, homologs of the nitric oxide receptor GUCY1 protein[40], were found to be expressed differently. While *GUCY1B1* had a low expression on the surface of redia and no visible expression in developing cercariae, it had low expression in the cerebral ganglia and strong surface expression in the tails, the main organ of locomotion in the freely swimming larvae (Fig. 5h). *GUCY1B2*, on the other hand, was expressed in two symmetrical pairs of neurons in the apical part of the body in developing cercaria inside redia and in the freely swimming larvae, but not in redia tissues (Fig. 5i).

## Discussion

Our data reveals that 14.5% of overall protein-coded genes were upregulated in rediae, while only 3.2% were upregulated in freely swimming cercariae. Similar stage-specific differences have been reported recently in other parasitic flatworms, such as *Fasciola gigantica*[41] and two species of psilostomatids[23]. Our GO analysis results partially support previous findings on model species *Schistosoma mansoni*, where cercariae was characterized as a stage with inactive cell division and upregulated genes involved in energy metabolism[42,43]. This similarity could be explained by the similar swimming strategy of *S. mansoni* and *C. lingua*, where both show intermittent swimming with active bursts and sinking phases believed to increase their lifespan and probability of infecting the next host[5]. While a few upregulated genes of certain neuronal families (GPCRs, Ion channels) were observed in cercaria of *S. mansoni*[42], these did not match the list of upregulated genes in dispersal larvae of *C. lingua*. Indeed, these larvae are not transcriptionally silent as was postulated for *S. mansoni*[42]: a large proportion of genes were constantly expressed among studied stages in *C. lingua*. Differences in the composition of upregulated genes raise questions about species-specific gene expression profiles related to differences in behavior cues and strategies for infecting the next host. These cues and strategies differ between mammal-infecting *S. mansoni* and fish-infecting *C. lingua*.

We believe that the dispersal larvae of trematodes exhibit a highly specialized functionality, primarily focused on energy store digestion (metabolism), motor coordination, and navigation. During the resource-rich feeding stage of the rediae, complex neuronal proteins, including ion channels, are synthesized, indicating the crucial role of this stage in the development of the nervous system in cercaria embryos.

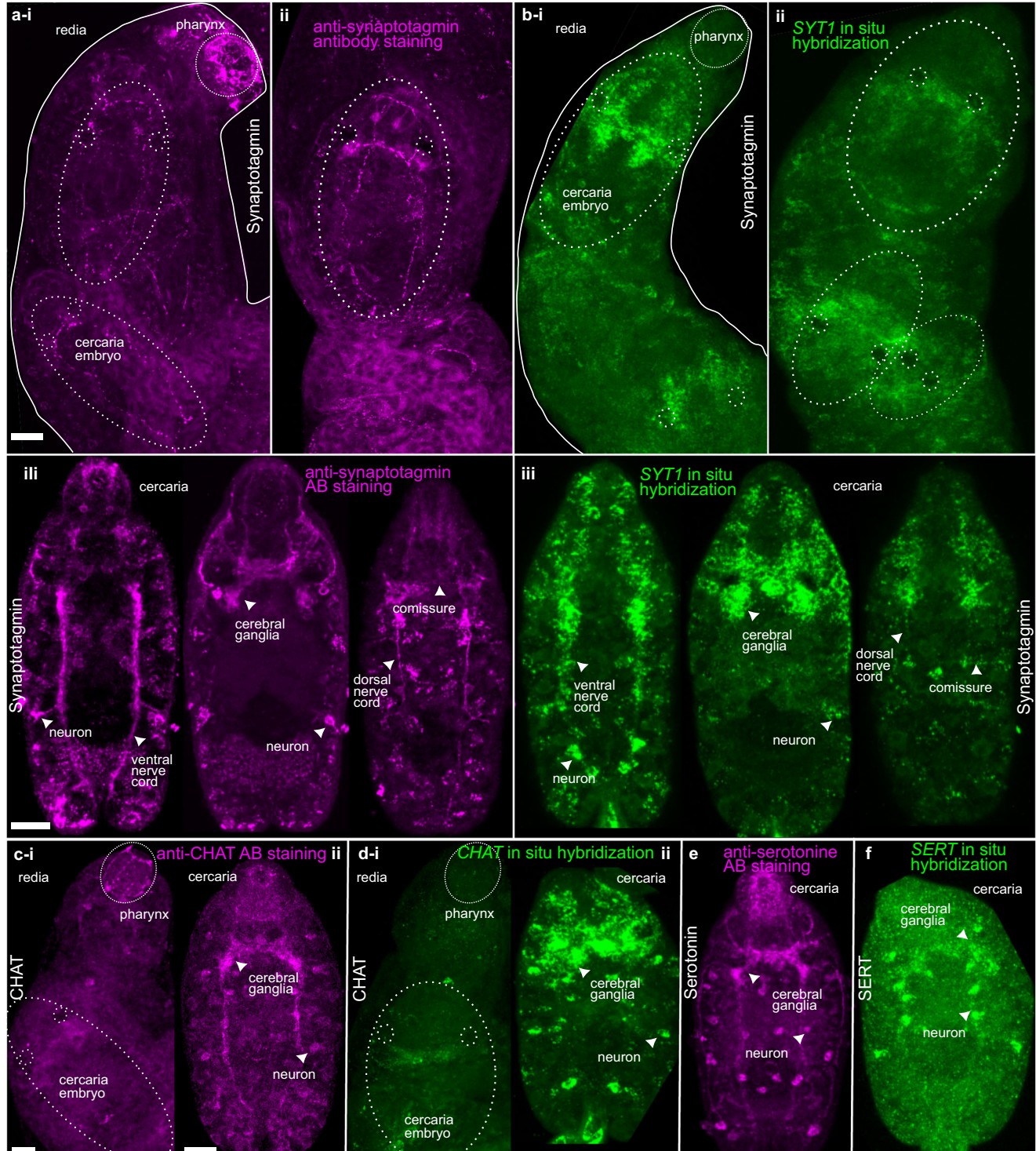

**Fig. 4 Parallel staining of neuronal markers by antibody staining and HCR in situ hybridization confirmed their localization in the nervous system.**
**a** Antibodies staining against SYT1 (magenta) and **b** HCR in situ hybridization of *Cl*SYT1 gene (green) in the nervous system of redia and developing cercaria inside (marked with dash circles) in proximal (**i**) and distal part of redia (**ii**) and optical sections of mature cercaria removed from the redia (**iii**).
**c** Antibodies staining against CHAT (magenta) and **d** HCR in situ hybridization of *Cl*CHAT gene (green) in the nervous system of redia and developing cercaria inside (marked with dash circles) (**i**) and in mature cercaria removed from the redia (**ii**). **e** Antibodies staining against serotonin (magenta) and **f** HCR in situ hybridization of *Cl*SLC6A4 (SERT) gene (green) in the nervous system of mature cercaria removed from the redia. Note signed parts of the central nervous system and peripheral neurons. Scale bar = 50 μm. Representative images were chosen from a pool comprising a minimum of 10 animals, randomly selected from three separate batches (infected mollusks) for each set of staining.

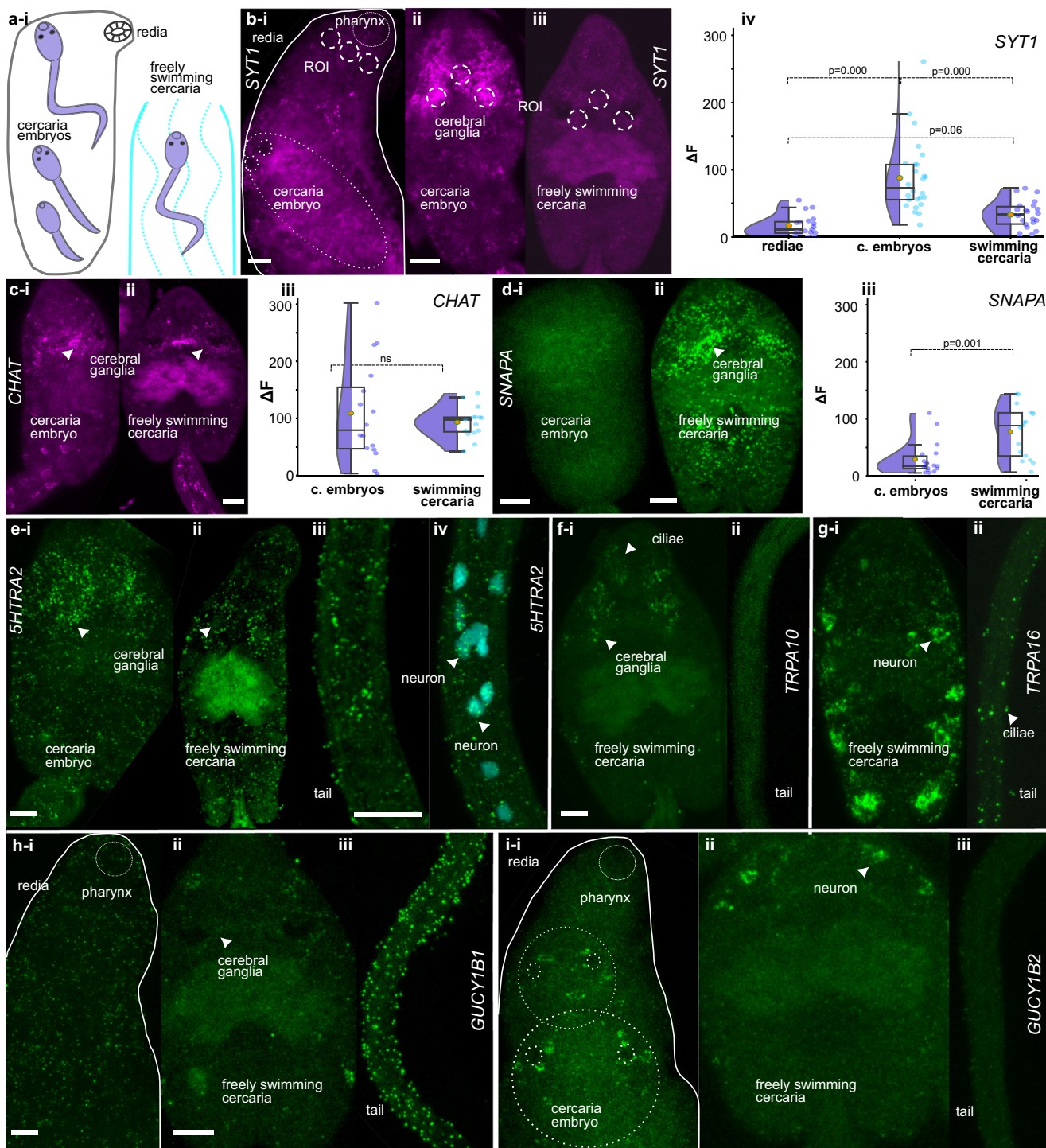

Our data on the expression of neuronal genes upregulated in dispersal cercariae included a few genes involved in synaptic vesicle trafficking and release. We found two TRPA channels that might be related to sensation, a putative GPCR receptor and two putative NO receptor soluble guanylate cyclase. A recent study showed active transcription in cercariae of *Schistosoma mansoni* localized in the tail region, including vesicle transport GO transcripts[44]. In contrast, in our data, expression of genes *ClSNAPA* and *ClPACSIN1*, involved in synaptic vesicle fusion and endocytosis, and GPCR *ClHTR1A2* was localized in the central nervous system and all over the body and tail of cercaria. For more details on TRPAs and soluble guanylate cyclases see below.

Our research has revealed the absence of gamma-aminobutyric acid (GABA) and histamine signaling pathways, as well as epinephrine and octopamine biosynthesis in *C. lingua*. These findings are consistent with previously published data based on genome analysis of a few parasitic flatworms[45], which have been discussed before[1,46,47]. Interestingly, despite the lack of genes for histamine biosynthesis and metabolism, one of the upregulated GPCR receptors in cercaria (*ClGPCR1*) is similar to the *S. mansoni* GPCR (with 29% similarity) that has been shown to have affinity for histamine[48]. Based on our transcriptome analysis, some special features of classical neurotransmitter pathways in *C. lingua* were observed. These include the similarity of

**Fig. 5 Localization of dispersal "stage-specific" genes in cercaria highlights their potential function. a** Stages studied (redia with developing cercaria and freely swimming cercaria). **b, c** Expression of neuronal markers (magenta) between stages and quantification of HCR fluorescence. **b** Synaptotagmin expression in redia (**i**), cercaria embryo (**ii**) and freely swimming cercaria (**iii**), and its quantification (**iv**). **c** *CHAT* expression in cercaria embryo (**i**) and freely swimming cercaria (**ii**), and its quantification (**iii**). **d–i** Expression of DEGs upregulated in dispersal cercaria stage. **d** *SNAPA* expression in cercaria embryo (**i**), and freely swimming cercaria (**ii**), and its quantification (**iii**). **e** *5HTRA2* gene expression in cercaria embryo (**i**), freely swimming cercaria body (**ii**), and its tail (**iii**), tail with nuclei stained with DAPI (**iv**). **f** *TRPA10* gene expression in freely swimming cercaria body (**i**), and tail (**ii**). **g** *TRPA16* gene expression in freely swimming cercaria body (**i**), and tail (**ii**). **h** *GUCYB1* gene expression in redia but not in cercaria embryos (**i**), freely swimming cercaria body (**ii**), and tail (**iii**). **i** *GUCYB2* gene expression in redia and in cercaria embryos (**i**), freely swimming cercaria body (**ii**), and no expression in the tail (**iii**). Note signed parts of the central nervous system and peripheral neurons. Scale bar = 50 μm. Representative images were chosen from a pool comprising a minimum of 10 animals, randomly selected from three separate batches (infected mollusks) for each set of staining. For quantitative comparisons, a minimum of 15 randomly selected animals from three separate batches were utilized for each group. The mean fluorescence intensity, after background subtraction, was compared, with each scatter point representing an individual animal. Test = Mann–Whitney *U* test or Kruskal–Wallis analysis of variance with Dunn's posthoc test. The box plots in **b**iv, **c**iii, and **d**iii depict the dataset's median (central line), mean (central point), interquartile range (box edges), and total data spread (whiskers).

| Taxa | | | Sodium VGS | GABA | Histamine | NOS |
|---|---|---|:---:|:---:|:---:|:---:|
| Spiralia | Platyhelminthes | free-living | 🟢 | 🟢 | 🟢 | 🔴 |
| | | *Trematoda* | 🔴 | 🔴 | 🔴 | 🔴 |
| | | *Cestoda* | 🟢 | 🔴 | 🟢 | 🔴 |
| | | *Monogenea* | 🟢 | 🔴 | 🔴 | 🔴 |
| | Lophotrochozoa | | 🟢 | 🟢 | 🟢 | 🟢 |
| | Gnatifera | | 🟢 | 🟢 | 🟢 | 🟢 |
| Ecdysozoa | Panarthropoda | | 🟢 | 🟢 | 🟢 | 🟢 |
| | *Nematoda* | | 🔴 | 🟢 | 🔴 | 🔴 |

Neurotransmitters pathways and channels. 🟢 exist 🔴 lost

**Fig. 6 The profiles of neuronal gene losses observed in Platyhelminthes exhibit parallels with those found in Nematoda across various groups within the Protostomia.** The taxa with the evolutionary transition from free-living to parasitic lifestyle are marked in bold. Data were compared across all published genomes for each group available at the time of publication (WormBase Parasite), in addition to three biological replicates of our dataset. VGS voltage-gated channels.

*Cl*SLC6A2 to both the Sodium-dependent dopamine transporter and the Sodium-dependent norepinephrine transporter, as well as the chloride channel nature of ionotropic acetylcholine receptors, as previously described in *S. mansoni*[47,49–52]. Moreover, the homologs of neuronal proteins identified in *C. lingua* are also similar to the genes involved in classical neurotransmitter pathways that have been functionally characterized in other trematode species[46,53–57].

Among the small gaseous messenger pathways in the nervous system, we found only orthologs involved in hydrogen sulfide biosynthesis in *C. lingua*. We discovered that *C. lingua* and 33 Platyhelminthes species, representatives of classes Trematoda, Monogenea, Cestoda, and free-living turbellarians *Macrostomum lignano* and *Schmidtea mediterranea* have lost the nitric oxide synthase (NOS) gene, which is essential for the cellular signaling NOS pathway[58]. We believe that two histochemical methods used to mark the presence of NOS in the tissues of parasitic flatworms based on antibody staining[59] and NADPH diaphorase histochemical staining[60] might have marked unspecific targets, as described earlier[61]. However, it is important to note that the genes encoding NOS exist in other lophotrochozoans[62,63] (Fig. 6). We propose that the loss of nitric oxide synthase in Platyhelminthes was compensated by the peculiarities of the parasitic lifestyle of these animals, which allows for unrestricted use of host-derived NO. Similar losses of the gene for this enzyme were reported for representatives of Ctenophora[64] and for nematodes in which it has been attributed to consequences of life in an environment enriched with nitric oxide by bacteria[65]. However, the lack of NOS in trematodes does not mean the absence of the NO signaling pathway. The upregulation of soluble guanylate cyclase in the *C. lingua* cercaria may indicate the involvement of NO in energy metabolism, motility, or adaptation to lower

concentrations of non-enzymatically produced NO[66] in seawater compared to host tissues. And in fact, we localized the expression of two studied soluble guanylate cyclases in both tail, the main organ of locomotion, and body neurons, suggesting its crucial involvement in the dispersal process and host detection (sensation). Nevertheless, the source of NO in cercariae outside the host body remains enigmatic and needs further study. Moreover, it was shown that bacterially derived NO enhances *C. elegans* longevity and stress resistance[65]. We speculate that NO deficiency may be a similar or even more critical reason for the short lifespan[8,67] of cercariae than energy limitations. It may have also played a crucial role in the evolution of digenean homo-, di-, and trixenic life cycles, where dropout of free-swimming larvae is common[20]. Additionally, the loss of NOS genes in both Platyhelminthes and Nematoda ancestors (Fig. 6) may have prompted the shift from a free-living to a parasitic lifestyle. To test this hypothesis, further studies are necessary.

The genes composition involved in synaptic vesicle cycle in bilaterians is conservative[24,27,62] and so as genes composition of synaptic vesicle cycle in *C. lingua*, which corresponds to recent identification of synaptic transmission machinery in different life cycle stage juveniles of liver fluke *Fasciola hepatica*[3].

Action potentials (APs) are a widespread signaling phenomenon in living organisms and in animals the fast AP is usually sodium-based. Sodium voltage-gated channels (SVGC) thought to have evolved alongside the early neuromuscular systems being evolutionary related to calcium voltage-gated channels[68,69]. The absence of sodium voltage-gated channels as in *C. lingua* was also described in *S. mansoni*[2] but not in other parasitic (Cestoda, Monogenea) and free-living flatworms clades[47,70–72]. Interestingly, we recorded APs in the tail of the *C. lingua* cercariae earlier[73]. The occurrence of APs in the absence of SVGC is

known for nematodes and is explained by the involvement of potassium and calcium VGC[74], which can be the case in *C. lingua* APs generation.

Cercaria of *C. lingua* has a complex behavior with characterized light and water turbulence perception playing a key role in finding the fish host[7,8,75,76]. The presence of rhabdomeric (r) Gq-opsins in *C. lingua*, as well as in *S. haematobium* and *S. mansoni*[30] along with previous reports of selectivity to different light spectra[8] suggests promising avenues for future experiments. Certain transient receptor potential ion (TRP) channels are associated with r-opsins and necessary for phototransduction[32]. Another TRPs can be involved in mechanotransduction[77] or thermosensation[39,78]. We suggested that at least some of TRP channels identified play a role in water turbulence perception and finding the fish host[79]. Cercariae *C. lingua* have multiple cilia at their surface[80] some of the long ciliae are shown to have connections to neurons[81]. In fact, two TRPA channels upregulated in cercaria were primarily expressed in the short cilia and in a set of peripheral neurons, highlighting their potential importance in the process of mechanosensation/transduction or heat/cold sensing ability and a next host finding.

The loss of neurotransmitter pathways is a prevalent occurrence observed across diverse taxa[62]. Within the Protostomia, the convergence between nematodes and trematodes highlights their shared absence of specific neurotransmitters and sodium voltage-gated channels (Fig. 6), which may imply a loss either before or after their specialization as parasites, or potentially a combination of both scenarios. Remarkably, the absence of nitric oxide synthases (NOS) in all taxa of Platyhelminthes, including both free-living and parasitic forms, strongly indicates the absence of NOS during an earlier phase in the evolutionary history of flatworms. It suggests a plausible evolutionary trajectory wherein the absence of particular neurotransmitter systems predates the specialization of proto-worms as parasites and can have a potential relationship with complex life cycles and other parasitic adaptations. Further investigation is necessary to understand how this combination of neurotransmitter systems interact, modulate neural activity, and contribute to the overall functionality of the nervous system.

## Methods

**Animal samples**. Cercariae and rediae of *Cryptocotyle lingua* (Creplin, 1825) (Digenea: Heterophyidae) were obtained from naturally infected common periwinkles *Littorina littorea* (Linnaeus, 1758), which were collected at the Biological Station of Zoological Institute RAS "Kartesh" (Chupa inlet, Kandalaksha bay, the White Sea) and in the vicinity of Bergen (Tyssøya, the North Sea). To collect the cercariae, each infected mollusk was placed individually in a reservoir filled with seawater of natural salinity (24‰). After four hours of cercariae emission, the larvae were collected and identified[82] under a dissection microscope. In order to clear the larvae sample of the mollusk mucus and other large xenogenic contaminations, cercariae were placed in the dark side of a sterile 100 mm Petri dish filled with sterile seawater (SSW: 22 μm filtered), where they migrated to the opposite side of the dish due to positive phototaxis. The cercaria were accumulated in 10 ml sterile tubes and cooled on thawing ice to 4 °C, which allowed the larvae to lose their mobility and be concentrated at the bottom of the tube. The original water was removed, and the larvae were rinsed with SSW in four passages. Each sample of living cercariae contained 600–800 individuals.

The rediae were recovered from the same mollusks after their dissection under a microscope and collected in a sterile Petri dish filled with SSW. Harvested rediae were immediately transferred to sterile round-bottom plastic wells for washing from scraps of host tissue and mucus. The washing with SSW was controlled with a thin pipet, carefully removing the remnants of host-derived tissues, cell aggregates, injured rediae, and cercariae released during the process. In general, each sample of live rediae (400–500 individuals) was washed 15 times in a total volume of 150 ml of SSW. To prevent adhesion of both rediae and cercariae to the surface of a sterile hydrophobic plastic, it was treated with a sterile solution of bovine serum albumin (BSA: 0.01 mg/ml). A part of the infected snail was kept in the aquarium facility of the Michael Sars Center, University of Bergen, and used for fixations and behavior experiments.

**RNA isolation and libraries preparation**. Total RNA isolation was performed using ExtractRNA™ (Evrogen, Moscow, Russia) according to the manufacturer's instructions, followed by intensive pipetting to fully lyse rediae and cercariae tissues. The RNA samples were stored in 80% EtOH at −80 °C for no longer than two weeks. Library construction for transcriptome sequencing was carried out using the TruSeq Stranded mRNA LT Sample Prep Kit from Illumina (San Diego, CA), following the manufacturer's protocol. We obtained three paired biological replicates for two life cycle stages: rediae and cercariae, from the same infected periwinkle in one pair. Sequencing of the six libraries was performed at Macrogen (Seoul, Korea) on an Illumina HiSeq 4000 platform with paired-end 2 × 100 reads.

**Species identification**. Morphological identification of species in digeneans can be challenging, even when using adult worms, and is sometimes impossible when using their larval stages[23,83]. To address this issue, we searched the transcriptome of *C. lingua* for the sequences of i) internal transcribed spacers 1 and 2 (ITS1 and ITS2), located between 18 S and 28 S in ribosomal RNA (rRNA), and ii) the mitochondrial cytochrome c oxidase subunit 1 gene (COX1). We found both complete sequences encoding 18S-ITS1-5.8S-ITS2-28S rRNA (GenBank: MW361240) and COX1 (GenBank: MW361241) to be 100% identical to the previously published sequences of *C. lingua*[84]. Thus, our morphological identification of the species, and the absence of cryptic species, is supported by molecular identification.

**De novo assembly and differential expression**. The quality of paired-end read data was assessed using FastQC (v0.11.9). Adapter and low-quality bases (Phred score below 30) were trimmed from sequence reads with Trimmomatic (v0.38)[85]. Sequences longer than 50 bp were retained for further analyses. The prepared paired-end read data from six libraries were pooled together and used in de novo assembly of the reference transcriptome using Trinity (v2.8.6)[86] with strand-specific read orientation and a minimum contig length of 200 bp. This assembly was further reduced by removing the low-expressed (FPKM < 3) and several xenogenic transcripts.

A local BLASTn approach was used to filter out potential xenogenic contaminating sequences from the raw assembly: with the host-mollusk *Littorina littorea* transcriptome of hemocytes and kidney (NCBI TSA: GGCG00000000.1)[87], whole body transcriptomes of related species *L. obtusata*, *L. fabalis*, *L. saxatilis*[88] and human genome (Encode: GRCh38, primary assembly). SILVA rRNA database[89] was also used to search for other potential xenogenic contaminations. The quality of the curated assembly was assessed by the presence of the Metazoa single-copy orthologues and verified using BUSCO (v4.1.4; metazoan-odb10)[90]. Only the longest proteins were selected as representatives of each gene for BUSCO benchmarking.

RNA-seq analysis was performed with Bowtie2 (v2.3.5.1)[91] and quantification method RSEM, to obtain gene expression values including counts, TPM (Transcripts Per Million), and TMM

(TMM-normalized TPM matrix). To identify differentially expressed genes (DEGs) in rediae and cercariae, an EdgeR test[92] for paired samples was carried out with the reference transcriptome and stage-specific reads in three replicates. Transcripts with absolute fold change (FC) values ≥ 4 and a false discovery rate (FDR)-corrected p-value ≤ 0.05 were considered as differentially expressed. To ensure that biologically meaningful differences were not overlooked, we conducted a detailed EdgeR analysis specifically on a subset of 300 verified neuronal genes, which includes the mentioned homeobox domain-containing proteins (see Supplementary Data 1). For this analysis, genes with an FDR ≤ 0.001 and FC > [1] were considered as differentially expressed.

**Amino-acid sequences prediction and functional annotation**. The coding regions within transcript sequences were identified using the TransDecoder tool (default criteria) with either BLASTp (Swissprot) or PFAM hits as ORF retention criteria. Transcripts encoding peptides shorter than 100 amino acids were excluded from the analysis. The predicted proteins were then annotated using the Trinotate tool with HMMER/PFAM protein domain identification, signalP (v4.1), TMHMM (v2.0c), and BLASTp (e-value threshold 1e-25) against the UniProt/SwissProt protein database. Additionally, protein families were annotated using InterProScan, and gene ontology (GO) mapping (e value cutoff 1e-25) was conducted in Blast2GO, including the curated transcriptome as the reference set. An enrichment GO analysis for the protein-coding DEGs (test set) was also performed in Blast2GO, and a Fisher's exact test was run with an FDR cut-off of 0.05. Only over-represented biological processes (BPs) were further analyzed.

**Identification of neuronal genes**. We conducted a similarity analysis and protein domain architecture study on neuronal genes, using reference proteins involved in neurotransmission, and sensory and neural pathways obtained from the Uniprot database. We primarily selected *Homo sapiens* proteins as reference species, but in cases where functionally characterized homologs were available, or if the reference proteins were absent in vertebrates, we used proteins from the trematode species *Schistosoma mansoni* or another invertebrate model species. To identify candidate homologs, we locally tBLASTn searched the *C. lingua* transcriptome with an E-value threshold of 1e-5 for short sequences. Any hits below this threshold were considered candidate homologs. We examined all candidate protein sequences for the presence of conserved amino-acid motifs related to Pfam[93] and SMART[94] domains. We used the InterProScan tool in Geneious R10 (https://www.geneious.com) to analyze protein domain architectures. We considered proteins whose domain content and order were identical to the corresponding prototypes as homologous. Supplementary Data 2–8 contain information on the reference genes, E-value of similarity, and NCBI accession numbers of homologous *C. lingua* transcripts. To validate the absence of anticipated genes, we conducted BLASTp and tBLASTn searches within the genomes of other Platyhelminthes and Nematoda species on WormBase Parasite v. WBPS14, contingent on the specific context (be it species, class, or phylum).

**Hybridization chain reaction (HCR) in situ hybridization**. Probes sets for HCR in situ hybridization were generated by insitu_probe_generator[95] and produced by Integrated DNA Technologies, Inc. (USA) for the following gene models: *ClCHAT*, *ClSYT1*, *ClSLC6A4* (SERT), *ClSNAP*, *ClPACSIN1*, *ClHTR1A2*, *ClGUCY1B2*, *ClGUCY1B1*, *ClTRPA16*, *ClTRPA10*, as well as for sense control sequences for *ClCHAT* and *ClGUCY1B2*. From 27

to 33 split initiator pairs were used for each gene (Supplementary Data 10).

Embryos were fixed in 4% formaldehyde overnight at 4ºC. Samples were then washed two times in phosphate-buffered saline and 0.1% Tween 20 (PBST). Samples were then dehydrated in 50% methanol and then with 100% methanol, and stored in 100% methanol at −20 °C until use.

The HCR protocol was based on a previously published protocol for fruit fly larva[36]. For a control experiment, we used negative control sense sequences and HCR reactions without probes. After HCR, animals were stained with DAPI and mounted on slides in 50% glycerol. All animals for HCR in situ hybridization were stained at once using the same hybridization, washing, and amplification time for all genes.

**Antibody staining**. For antibody staining, animals were fixed in 4% formaldehyde overnight and kept in PBST at 4 °C. We used commercial antibodies, including anti-serotonin rabbit, anti-synaptotagmin rabbit (Sigma), anti-CHAT goat (Invitrogen), Alexa Fluor 488 donkey anti-rabbit, and Alexa Fluor 555 donkey anti-goat (Molecular Probes). Primary antibodies were a kind gift from Prof. Dr. Andreas Hejnol, Friedrich-Schiller-Universität Jena. Prior to staining, animals were treated with Proteinase K and incubated overnight in a blocking solution. We stained animals with primary and secondary antibodies for 5 days at a 1:800 dilution in PBST containing 1% BSA (Sigma). After washing in PBST, animals were stained with DAPI and mounted on slides in 50% glycerol.

**Imaging**. HCR in situ and antibody-stained samples were imaged using Olympus fv3000 and Leica Stellaris 8 microscopes using a ×40 objectives. Z-sections of whole animals were taken with the same laser power and settings for all HCR in situ animals.

**Quantification of HCR in situ signals**. For each gene, the mean fluorescence intensity of three identical diameter circular regions of interest (ROI) was measured in ImageJ (NIH) for the two cerebral ganglia and a cerebral commissure at the Z-section of the animal. The background signal was measured for the random region within the animal outside the central nervous system localization and subtracted from the mean fluorescence of three ROIs.

**Behavior recording and video analysis**. We used the behavior setup described earlier[22]. Utilizing a 3D printed PLA mold, modeled similar to what was described earlier[22] but with a reduced height to facilitate the 2D tracking of small cercaria, we fabricated single-use agarose arenas (1% in Artificial Sea Water (ASW)) that were 10 mm in diameter and 1 mm high (approximate volume 78.5 mm³). In the setup, the arena was encircled by a PLA ring embedded with infrared (IR, peak emission 850 nm) LEDs and was shielded from external light by a non-transparent plastic cover (tube).

Videos were recorded under controlled temperature conditions at 15 °C using an IR-sensitive monochrome camera (DMK 33UP1300, The Imaging Source, Germany) and IC Capture software, operated through an Arduino-based circuit and interfaced with a GUI written in Python[22]. For recording, we positioned 1 animal in an agarose arena and allowed a 1-minute acclimatization period in the dark. We then recorded the animals for 5 minutes in the absence of visible light with 30 frames/s.

Following acquisition, the videos were analyzed using the program ToxTrac[21] with the default settings, which tracked the position of the animal by identifying the center of its detected shape. Prior to analysis, we inverted all video frames using

VirtualDub software. Cercariae were recorded four hours post-emission. We utilized ten larvae from three different mollusk hosts. After dissection, we also recorded ten rediae, mostly small non-gravid individuals, under identical conditions in ASW. Positional data, calibrated against the arena and pixel size, were used for the computation of movement trajectories, from which we derived the speeds and path complexities.

Speed calculations were based on trajectory data sampled every fifth frame. We computed Euclidean distances between consecutive points[22], which, divided by the elapsed time between frames, yielded speed values. These data were smoothed with a running average over three frames. Only animals with smoothed speeds exceeding 200 μm/s were categorized as active, to exclude cercariae drift. This activity threshold determined the timing of different behavioral states—swimming versus quiescence or floating. One non-motile larva was excluded from the analysis. The speed distribution and activity timing for the remaining nine active cercariae are displayed in Fig. 1c and Supplementary Fig. 1.

Local path complexity was quantified by embedding matrices of (x,y) positions within 30-second windows, applying entropy calculations using functions published earlier[22]. The singular value decomposition of these matrices allowed us to compute the entropy of the normalized eigenvalues, where lower complexity indicates straighter paths, and higher complexity reflects more varied speeds and directions. Since this measure is unaffected by the mean position, speed, or orientation, it uniquely quantifies the variability of an animal's trajectory within the specific interval[22]. The distribution of these path complexity values is shown in Supplementary Fig. 1.

**Data visualization and analysis**. All data analysis and visualization were performed with Python 3.11 using the numpy, pandas, scipy, scikit-learn, matplotlib, and seaborn libraries.

**Statistics and reproducibility**. In assessing sequencing data, three randomly chosen infected mollusks were subjected to cercarial shedding, followed by the dissection of rediae. Behavioral studies, in situ hybridization, and immunostaining procedures were conducted using a combined sample pool derived from hundreds of cercariae and rediae obtained from at least three different infected mollusks. The selection of individual animals for sampling was random, with each experimental set comprising at least ten randomly chosen individuals. The specific sample sizes used for quantifications and comparisons across different conditions are detailed within the figure legends. Samples were checked by Shapiro–Wilk normality test and compared with Kruskal–Wallis analysis of variance with Dunn's posthoc test and Mann–Whitney U tests.

**Reporting summary**. Further information on research design is available in the Nature Portfolio Reporting Summary linked to this article.

## Data availability

The data that support the findings of this study are openly available in GenBank and Figshare. The raw data have been deposited in DDBJ/EMBL/GenBank (Bio-Project: PRJNA646866; BioSample: SAMN15567434, SAMN15567352; SRA accession: SRR12246812 - SRR12246817). The curated *C. lingua* Transcriptome Shotgun Assembly (TSA) has been deposited at DDBJ/EMBL/GenBank under the accession GISJ00000000. Both, raw and curated (v1.2.2) assemblies are available at Figshare (https://doi.org/10.6084/m9.figshare.12673415.v1; https://doi.org/10.6084/m9.figshare.13386671.v1; https://doi.org/10.6084/m9.figshare.13387091.v1). The source data used in the figures are provided in Supplementary Data 11.

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

## Acknowledgements

We are grateful to Dr Alexander Klimovitch and Dr Kirill Nikolaev for their kind help in initial sampling. M.C. and O.T. were supported by two grants from the Research Council of Norway: grant number 339399 to M.C. and grant number 234817 to Sars International Center for Marine Molecular Biology Research (supporting both M.C. and O.T.). A.G. was supported by the IEPhB Research Program 075-0152-22-00 and IEPhB Research Resource Center. We thank Mie Wong for comments on the manuscript.

## Author contributions

Conceptualization: O.T., A.G., M.C. Methodology: O.T., A.G., M.C. Investigation: O.T., A.G. Visualization: O.T., A.G. Funding acquisition and administration: M.C. Writing: O.T., A.G., and M.C.

## Funding

## Competing interests

The authors declare no competing interests.
