## [Peer Review File · Communications Biology]

Reviewers' comments:

Reviewer #1 (Remarks to the Author):

In the manuscript by Tolstenkov the authors describe the molecular workings of the nervous system in *Cryptocotyle lingua*. The findings show increased levels of certain neuronal genes in the dispersal larvae, particularly those linked to synaptic vesicle trafficking, TRPA channels, and nitric oxide receptors. The authors suggest that the findings have functional relevance to larval adaptations and host-finding strategies. The results and the related datasets shared by the authors improve knowledge on trematode neurobiology and will be of interest for the community.

Comments to the authors:

- 1) EdgeR is expected to yield better performance and greater statistical power compared to a t-test, so I do not understand why the authors decided to pursue this approach. Moreover, it remains unclear which analysis in the manuscript employs this test.
- 2) Authors should be more careful when extrapolating the absence of transcripts in the de novo assembly to the absence of the gene in the organism, as this method has biological and technical limitations to reveal all genes present in a species.
- 3) Specify the parameters used to classify a gene as coding or non-coding.
- 4) How were mobile elements identified?
- 5) Line 88 change Fig 2D and E by Fig 1D and E
- 6) Line 117 - The deduction concerning homeobox genes cannot be inferred from Fig 2B. Given that the comprehensive list of differentially expressed genes isn't displayed, such a conclusion cannot be drawn from the provided data. Kindly consider adding a supplementary table that encompasses all DE genes along with their corresponding statistics.
- 7) Line 150 - When you mention conducting differential expression analysis on specific genes, does this imply re-running the EdgeR analysis using raw counts solely for these genes, or does it refer to extracting the results only for this gene family from the overall table?
- 8) Line 169 - looking at Fig 3B in some categories the percentage of regulated genes is more than 50% so this statement should be clarified.
- 9) Line 181 - the is no figure 3C (probably this refers to Fig 3A III)
- 10) What statistical measure does the e-value presented in the supplementary tables correspond to? Is it the blast e-value?
- 11) Supplementary Figure 5 - comments in column H are not in English.
- 12) Line 225 - Supplementary Figure 5 does not show expression levels.
- 13) Line 248 - change to S Figure 2B
- 14) Line 253 - change to S Figure 2C
- 15) Line 261 and 269 - change to S Figure 2D
- 16) "S." is a more standard abbreviation for the *Schistosoma* genus than "Sch."

Reviewer #2 (Remarks to the Author):

Using transcriptional analysis and in situ hybridization, the investigators have made a comprehensive accounting for differences in molecular expression between two distinct developmental stages of *Cryptocotyle lingua*. There is a strong argument that we need to broaden the number of model systems with simple nervous systems for future study. In addition, there is a large gap in our understanding of how drastic life cycle changes are controlled at the genetic / molecular level as demonstrated in *C. lingua*. The paper for the most part presents the results clearly and I believe it is a comprehensive and important contribution in the establishment of *C. lingua* as a model system.

The results are technically sound, but I have some comments on the brief "behavioural" component of the paper. The abstract states that "Additionally, employing a behavior quantification toolkit, we assessed cercaria motility, facilitating further investigations in the behavior and physiology of parasitic flatworms." However the actual results on behavior are reported only as a few panels in the first figure and some panels show just a single trajectory from I assume a single animal. Thus the interpretation of the results is more subjective than objective, given the limited information we have from the panels and the methods sections. This needs to be addressed. Should a statement important enough to be included in the abstract be supported with such a limited amount of data?

Even though there is a reference, the "complexity" measure should be defined in the material and methods.

Fig 1 IV. Using "min" and "max" and an arbitrary heat map, for complexity makes it impossible to judge the amount of variation in the behavior during the activity "bursts." It would be more clear if we could see the distribution of complexity measures.

FIG 1 V. Only 3 bursts from a single path is shown here. How has this path been selected? Is this representative? What about other trajectories from other samples?

RESPONSE TO REVIEWERS (responses are shown in bold italic)

Reviewer #1 (Remarks to the Author):

In the manuscript by Tolstenkov the authors describe the molecular workings of the nervous system in *Cryptocotyle lingua*. The findings show increased levels of certain neuronal genes in the dispersal larvae, particularly those linked to synaptic vesicle trafficking, TRPA channels, and nitric oxide receptors. The authors suggest that the findings have functional relevance to larval adaptations and host-finding strategies. The results and the related datasets shared by the authors improve knowledge on trematode neurobiology and will be of interest for the community.

Response: We appreciate the reviewer's positive feedback and acknowledgment of our work's contribution to flatworm neuroscience. Indeed, we believe that the comprehensive dataset on neuronal genes we have generated is of significant value for broader applications in the field.

Comments to the authors:

1) EdgeR is expected to yield better performance and greater statistical power compared to a t-test, so I do not understand why the authors decided to pursue this approach. Moreover, it remains unclear which analysis in the manuscript employs this test.

We acknowledge the reviewer's concern regarding the use of the T-test. In response, we expanded our EdgeR analysis to not only encompass the general characterization of the transcriptome (as originally done) but also a targeted analysis of the specific genes described. This subset comprises 74 homeoboxes and 226 characterized neuronal genes, detailed in the updated supplementary table 1-9. While this revised analysis refined our list of DEGs, the overall conclusions of the paper remain unchanged. We've incorporated these changes in the text and updated Figure 3, as well as Supplement figures 2 and 3 (Please note that we added a Supplementary figure 1 in response to the comments of the Reviewer 2 that caused a shift of Supplementary figures numbers).

Revisions Made:

Line 444 (Please note that the line numbers are corresponded to the revised manuscript): We've clarified our approach in Methods: "To ensure that biologically significant differences were not overlooked, we conducted a detailed EdgeR analysis specifically on a subset of 300 verified neuronal genes, which includes the mentioned homeobox domain-containing proteins (see Supplementary Table 1). For this analysis, genes with an $FDR \leq 0.001$ and $FC > [1]$ were considered as differentially expressed." Please note that adding a Supplementary Table 1 shifted the Supplement Table numbers.

Line 98: Additional details have been provided for context: "The primary biological function of the redia stage is the production of dispersal larvae, the cercariae. Accordingly, of the 74 development regulatory Homeobox genes, only six were upregulated in the cercariae. In

Supplement Figure 2.

Supplement Figure 3.

In Supplement Figures 2 and 3 we updated heatmaps and figure legends to include following statement: Differential expression = $FDR \leq 0.001$ and $FC > [1]$, green apertyx – genes upregulated in redia, magenta apertyx – genes upregulated in cercaria.

Please also see updated Supplementary Tables 1 - 9.

2) Authors should be more careful when extrapolating the absence of transcripts in the de novo assembly to the absence of the gene in the organism, as this method has biological and technical limitations to reveal all genes present in a species.

We completely agree with your point. Not all genes can be expressed at the developmental stages we studied. Therefore, all cases of expected but missing genes were confirmed by blast searches in the genomes of the other 25 trematode species (WormBase Parasite). This was stated directly in the Results for the case of NO-synthase (lines 148-150) and has now been clarified in Methods (line 472).

"To validate the absence of anticipated genes, we conducted BLASTp and tBLASTn searches within the genomes of other Platyhelminthes and Nematoda species on WormBase Parasite v. WBPS14, contingent on the specific context (be it species, class, or phylum)."

3) Specify the parameters used to classify a gene as coding or non-coding.

Thank you for highlighting this aspect. As previously outlined in the Methods section (line 449), we've provided additional clarity on our methodology:

"The coding sequences were identified using the TransDecoder tool (default criteria) with either BLASTp (Swissprot) and PFAM hits as ORF retention criteria. Transcripts encoding peptides shorter than 100 amino acids were excluded from the analysis."

4) How were mobile elements identified?

We appreciate the attention to this detail. Indeed, the prevalence of expressed transposons and other mobile elements is often overlooked in protostomian transcriptome studies. Although this topic is beyond the primary focus of our current paper, we felt it was necessary to address it briefly. We are in the process of conducting a detailed analysis of mobile elements within the C. lingua transcriptome and aim to present our findings in a subsequent publication.

To identify these elements in the current study, we relied on several criteria. These included the presence of Pfam domains characteristic of various families of reverse transcriptases, transposases, and integrases. Furthermore, a significant similarity based on blastp searches (with a threshold of $1.0E < 10$) to prototype transposons (e.g., LINE-1, PiggyBac, Tigger, Pogo) and retrovirus-like hits was also used to classify certain proteins as derivatives of transposable elements.

We have now added clarity in the manuscript (line 93):

"Our C. lingua transcriptome analysis revealed a high number of mobile elements such as LINE-1, PiggyBac, Tigger, Pogo transposable elements, and gag-pol polyproteins. However, for the scope of this study, these elements were excluded from our primary analyses."

5) Line 88 change Fig 2D and E by Fig 1D and E

Corrected.

6) Line 117 - The deduction concerning homeobox genes cannot be inferred from Fig 2B. Given that the comprehensive list of differentially expressed genes isn't displayed, such a conclusion cannot be drawn from the provided data. Kindly consider adding a supplementary table that encompasses all DE genes along with their corresponding statistics.

Thank you for the constructive feedback. We recognize the importance of providing comprehensive data to support our conclusions regarding homeobox genes. In response, we have incorporated a new table detailing Homeobox domain-containing proteins as Supplementary table 1. This table, along with others, now includes results from the specific EdgeR analysis accompanied by the relevant statistics. Please refer to our response to comment 1 for a detailed explanation and the associated text corrections.

7) Line 150 – When you mention conducting differential expression analysis on specific genes, does this imply re-running the EdgeR analysis using raw counts solely for these genes, or does it refer to extracting the results only for this gene family from the overall table?

For the 300 genes specifically characterized in the paper, we implied re-running the EdgeR analysis using raw counts solely for these genes. Please refer to our response to comment 1 for a detailed explanation and the associated text corrections.

8) Line 169 - looking at Fig 3B in some categories the percentage of regulated genes is more than 50% so this statement should be clarified.

Corrected consistent with the new DEG results after specific subset of genes EdgeR analysis (see also updated Figure 3B and Supplemental Table 2).

9) Line 181 – there is no figure 3C (probably this refers to Fig 3A III)

Corrected (complete list of the enzymes can be found in the Supplemental information).

10) What statistical measure does the e-value presented in the supplementary tables correspond to? Is it the blast e-value?

This is blastp e-value. Now it is clarified in all supplementary tables.

11) Supplementary Figure 5 - comments in column H are not in English.

Corrected in the Supplement Table 6 (former 5).

12) Line 225 - Supplementary Figure 5 does not show expression levels.

We could not identify Supplementary Figure 5. Regarding the Supplement Tables and the Supplement Figures 1 and 2, they were updated.

13) Line 248 - change to S Figure 2B

Addressed.

14) Line 253 - change to S Figure 2C

Corrected.

15) Line 261 and 269 - change to S Figure 2D

Addressed.

16) "S." is a more standard abbreviation for the Schistosoma genus than "Sch."

Addressed.

Reviewer #2 (Remarks to the Author):

Using transcriptional analysis and in situ hybridization, the investigators have made a comprehensive accounting for differences in molecular expression between two distinct developmental stages of *Cryptocotyle lingua*. There is a strong argument that we need to broaden the number of model systems with simple nervous systems for future study. In addition, there is a large gap in our understanding of how drastic life cycle changes are controlled at the genetic / molecular level as demonstrated in *C. lingua*. The paper for the most part presents the results clearly and I believe it is a comprehensive and important contribution in the establishment of *C. lingua* as a model system.

The results are technically sound, but I have some comments on the brief "behavioural" component of the paper. The abstract states that "Additionally, employing a behavior quantification toolkit, we assessed cercaria motility, facilitating further investigations in the behavior and physiology of parasitic flatworms." However the actual results on behavior are reported only as a few panels in the first figure and some panels show just a single trajectory

from I assume a single animal. Thus the interpretation of the results is more subjective than objective, given the limited information we have from the panels and the methods sections. This needs to be addressed. Should a statement important enough to be included in the abstract be supported with such a limited amount of data?

*We deeply appreciate the reviewer's constructive feedback and positive remarks on the our efforts towards establishing *Cryptocotyle lingua* as a model system and presenting a few toolkits for the community.*

We recognize the presented behavioral data might have been too brief, not fully representing its significance as indicated in the abstract. This was a result of our aim for conciseness of the text which we agreed was not too successful.

To balance this inequality we have introduced Supplementary Figure 1, which encompasses the distribution of speed, complexity, activity timings, and extra trajectories for the sampled animals.

From the batch derived from several infected mollusks, 10 random individual larvae were recorded, with one trajectory being unsuccessful. Our primary goal was to illustrate the toolkit's capabilities rather than provide an exhaustive behavioral analysis, which was not the main scope of the manuscript, explaining the limited sample size.

Supplementary figure 1. Visualization of the trajectory complexity and the burst of speed for each sampled freely moving cercaria larvae. Note that animal 3 is plotted in the main figures (Figure 1 C). **(A)** The trajectories are color-coded to represent the level of complexity (entropy) in the path of specific animals. **(B)** Speed distribution histogram for the sampled animals. **(C)** Timing of activity states for the sampled animals with the floating/ swimming threshold at 200 micrometers/second. **(D)** Normalized complexity distribution histogram for the same sampled animals.

Even though there is a reference, the "complexity" measure should be defined in the material and methods.

A clearer definition and explanation of the "complexity" measure is now included in the Methods section.

Both the Results and Methods sections have undergone revisions to more comprehensively detail the behavioral component. We have to keep the main text within 5000 words limits and therefore we significantly increased the methods section.

Line 74. Results section. "We characterized speed, timing of activity states and path complexity of a limited sample of cercaria using computer vision and analysis toolkit^{24,25} (see methods section for details)".

Line 508. Methods section. "We used behavior setup described earlier²². Utilizing a 3D printed PLA mould, modeled similar to what was described earlier²² but with a reduced height to facilitate the 2D tracking of small cercaria we fabricated single-use agarose arenas (1% in Artificial Sea Water (ASW)) that were 10 mm in diameter and 1 mm high (approximate volume 78.5 mm³). In the setup, the arena was encircled by a PLA ring embedded with infrared (IR, peak emission 850 nm) LEDs and was shielded from external light by a non-transparent plastic cover (tube).

Videos were recorded under controlled temperature conditions at 15° C using an IR-sensitive monochrome camera (DMK 33UP1300, The Imaging Source, Germany) and IC Capture software, operated through an Arduino-based circuit and interfaced with a GUI written in Python²². For recording, we positioned 1 animal in an agarose arena and allowed a 1-minute acclimatization period in the dark. We then recorded the animals for 5 minutes in the absence of visible light with 30 frames /s.

Following acquisition, the videos were analyzed using the program ToxTrac²¹ with the default settings, which tracked the position of the animal by identifying the centre of its detected shape. Before analysis, all frames for each video were inverted using Virtual Dub software. We recorded the cercaria after four hours after emission. In total, ten larvae from three mollusk hosts were used. Subsequently, we dissected the mollusks, and ten rediae of various sizes, primarily consisting of small non-gravid rediae, were recorded under the same conditions in ASW. All acquired positional data were calibrated based on the size of the arena and calculated pixel size to derive movement trajectories. This data was then used to compute the speed and path complexity of the animal trajectories.

We sampled trajectory data every 5th frame for speed calculations. Euclidean distance between consecutive points was computed²², which, when divided by the time between sampled frames, provided speed. Speed data was smoothed using a running average window of three. Animals showing smoothed speeds over 200 µm/s were considered active to exclude cercariae drift. Using this threshold and the duration in each state (swimming or quiescence and floating), we determined the timing of activity states. Animal not moving (1 out of 10) was omitted. We

present the distribution of speed values and timing of nine actively swimming cercaria in Supplementary Figure 1.

Local path complexity was quantified using embedding matrices of positions within a specific time frame, from which entropy-based local path complexity is determined using functions published earlier²². For each time window, 30 s for our analysis, we examined a matrix of (x,y) positions over all frames, centering on position variations around the average within the chosen timeframe. After performing singular value decomposition on this matrix, we computed the entropy of normalized eigenvalues. Lower complexity values from this method signify predictable, straight-line paths, while varied speeds and movement directions result in higher values. This metric remains unaffected by absolute mean position, speed, and orientation in the timeframe, hence local path complexity solely represents trace variability during a specific interval, independent of average movement speed²². We present the distribution of path complexity values in Supplementary Figure 1."

Fig 1 IV. Using "min" and "max" and an arbitrary heat map, for complexity makes it impossible to judge the amount of variation in the behavior during the activity "bursts." It would be more clear if we could see the distribution of complexity measures.

For Figure 1 IV, we've supplemented the data with distribution histogram charts in Supplementary Figure 1, offering a clearer insight into the speed and complexity measures during activity bursts.

Fig 1 V. Only 3 bursts from a single path is shown here. How has this path been selected? Is this representative? What about other trajectories from other samples?

Concerning Figure 1 V, we initially showcased specific bursts as illustrative and representative samples. To provide a broader perspective, we've now added more trajectories from various sampled animals to the Supplement Figure 1.

We believe these modifications adequately address the concerns raised, and we are appreciative of the constructive feedback, which has allowed us to refine our manuscript.

REVIEWERS' COMMENTS:

Reviewer #1 (Remarks to the Author):

I appreciate the author's modifications in response to the suggestions and have no additional comments.

Reviewer #2 (Remarks to the Author):

The manuscript is much improved and I am happy with the changes made in response to my comments.

minor:

Please double check for typos and grammar for the added text in the M&M. E.g.

"We used behavior setup described earlier 22"

RESPONSE TO REVIEWERS (responses are shown in bold italic)

Reviewer #2 (Remarks to the Author):

The manuscript is much improved and I am happy with the changes made in response to my comments.

minor:

Please double check for typos and grammar for the added text in the M&M. E.g.

"We used behavior setup described earlier ²²"

Thank you for pointing out minor typos! We corrected added text:

Behavior recording and video analysis

We used behavior setup described earlier ²². Utilizing a 3D printed PLA mold, modeled similar to what was described earlier ²² but with a reduced height to facilitate the 2D tracking of small cercaria, we fabricated single-use agarose arenas (1% in Artificial Sea Water (ASW)) that were 10 mm in diameter and 1 mm high (approximate volume 78.5 mm³). In the setup, the arena was encircled by a PLA ring embedded with infrared (IR, peak emission 850 nm) LEDs and was shielded from external light by a non-transparent plastic cover (tube).

Videos were recorded under controlled temperature conditions at 15° C using an IR-sensitive monochrome camera (DMK 33UP1300, The Imaging Source, Germany) and IC Capture software, operated through an Arduino-based circuit and interfaced with a GUI written in Python ²². For recording, we positioned 1 animal in an agarose arena and allowed a 1-minute acclimatization period in the dark. We then recorded the animals for 5 minutes in the absence of visible light with 30 frames /s.

Following acquisition, the videos were analyzed using the program ToxTrac ²¹ with the default settings, which tracked the position of the animal by identifying the centre of its detected

shape. Prior to analysis, we inverted all video frames using VirtualDub software. Cercariae were recorded four hours post-emission. We utilized ten larvae from three different mollusk hosts. After dissection, we also recorded ten rediae, mostly small non-gravid individuals, under identical conditions in ASW. Positional data, calibrated against the arena and pixel size, were used for the computation of movement trajectories, from which we derived the speeds and path complexities.

Speed calculations were based on trajectory data sampled every fifth frame. We computed Euclidean distances between consecutive points²², which, divided by the elapsed time between frames, yielded speed values. These data were smoothed with a running average over three frames. Only animals with smoothed speeds exceeding 200 $\mu\text{m/s}$ were categorized as active, to exclude cercariae drift. This activity threshold determined the timing of different behavioral states—swimming versus quiescence or floating. One non-motile larva was excluded from the analysis. The speed distribution and activity timing for the remaining nine active cercariae are displayed in Fig. 1c and Supplementary Fig. 1.

Local path complexity was quantified by embedding matrices of (x,y) positions within 30-second windows, applying entropy calculations using functions published earlier²². Singular value decomposition of these matrices allowed us to compute the entropy of the normalized eigenvalues, where lower complexity indicates straighter paths, and higher complexity reflects more varied speeds and directions. Since this measure is unaffected by the mean position, speed, or orientation, it uniquely quantifies the variability of an animal's trajectory within the specific interval²². The distribution of these path complexity values is shown in Supplementary Fig. 1.